# Scientific and engineering practices aligned with the NGSS in the performance of secondary stage physics teachers

Mohammad Khair M. Al-Salamat [ID] *

Department of Curriculum and Educational Technology, College of Education, Taif University, Taif, Saudi Arabia

* m.alsalamat@tu.edu.sa

## Abstract

The study aimed to unravel the degree to which scientific and engineering practices complied with the NGSS (Next Generation Science Standards) were present in the performance of physics teachers at the secondary stage in Taif city, Saudi Arabia. The study also sought to identify the effect of the qualification and years of experience variables on the degree to which the NGSS-aligned practices were present in the performance of physics teachers. The study adopted a mixed method in applying a closed-ended questionnaire and an interview. The questionnaire comprised 44 indicators and was applied to (49) teachers who were randomly selected. The interview was applied to (6) education supervisors who were purposively selected. The results revealed that the following five practices: "asking questions and identifying problems"; "obtaining, evaluating, and communicating information"; "planning and carrying out investigations"; "analyzing and interpreting data"; and "involvement with proofs and evidence" all rated medium. The three practices, "interpretation and solutions design"; "developing and using models"; and "using mathematics and computational thinking" all rated weak. Data analysis of education supervisors' interviews revealed that their opinions were, to an extent, concordant with the questionnaire findings. The results also revealed that graduate teachers exhibited a higher level of these practices than the teachers with just BS degrees while years of experience, as a variable, had no significant difference in teachers' use of these practices. It was concluded that teachers of physics need to reconsider their teaching practices to be aligned with NGSS. This study can be a contribution to teachers' professional development efforts that target the alignment of science instruction with the NGSS.

**Data Availability Statement:** All relevant data are within the paper and its Supporting Information files.

## Introduction and literature review

Educational standards are one of the most prominent issues in the field of education in general, and evaluation in particular, due to the acceptance and interaction they received from those involved in it. Education and teaching institutions adopted such standards in order to develop their education systems because of the active role these standards play in accurately

**Funding:** yes Taif University The funders had no role in study design, data collection and analysis, decision to publish, or preparation of the manuscript.

**Competing interests:** The authors have declared that no competing interests exist.

identifying the targeted outcomes of the education process [1–4]. Several institutions aimed to develop the field of teaching and learning science based on these standards. Among these institutions, there are the National Science Teaching Association (NSTA), the National Service Framework (NSF), and the American Association for the Advancement of Sciences (AAAS). Many projects for developing science teaching appeared, such as the Specialized Technical Services (STS) for reforming science curricula in light of the interaction between science, technology, and society; the National Standards for Teaching Sciences; the 2061 Project; and the California Standards for Science Curriculum.

In addition to the preceding projects, a new document for science standards was developed in 2011 by twenty-six American States under the auspices of the National Research Council (NRC) and a team of (41) members. The document dealt with science for all stages, from kindergarten till grade twelve (K-12) [5]. This document ultimately produced the Next Generation Science Standards (NGSS) which were considered to be the most updated approach in reforming and developing the teaching and learning of science. The NGSS were designed to suit all classes from kindergarten to high school (K-12). They were also meant to anticipate the future performance of students. The NGSS have three major dimensions: main ideas, scientific and engineering practices, and comprehensive concepts [6, 7]. The NGSS of 2013 are seen as a clear progression from the previous standards [8] in perceiving scientific knowledge as well as the relationship between scientific concepts and their practices. Such conceptions necessitate updating how science teaching is organized and delivered [9, 10].

According to Zaitoon [11], the objectives of students' learning of sciences in general and physics in particular have changed; therefore, science teachers of all levels need to adapt themselves to these new standards in order to achieve an actual change in their teaching practices. Therefore, educators believe it is essential to develop teachers qualifying programs that help teachers adjust their teaching practices and provide them with training opportunities which target the NGSS. Teachers' conceptions about the NGSS practices should also be reshaped in a way that contributes to the achievement of the required NGSS objectives [12, 13].

Zaitoon [11] claimed that among the most important objectives of science teaching is making students behave and practice in a scholarly manner to gain the skills of engineering design which in turn enables them to conduct research and to solve the problems they encounter while studying science or in real life. The NGSS adopted the term "scientific practices" instead of science processes because it combines the practices and skills of the scientist, i.e. whoever teaches science, in addition to the skills of the engineer who solves problems [14]. This is to ensure that students acquire the correct and real mechanism by which scientific research is conducted. It is the mechanism through which scientists, like Einstein, were able to come up with their discoveries [8, 13, 15]. Scientific and engineering practices can be described as "cognitive, discursive, and social activities" that can be utilized in science teaching to achieve a profound understanding of the very nature of science. They include building and using models, constructing logical arguments, tackling questions, and providing evidence-based explanations of natural phenomena [16].

The official website of the NGSS and various studies such as [2–4, 15, 17–20] tackled scientific and engineering practices from several perspectives: whether to measure their range of realization in teaching, to reveal the attitudes of the people involved, or to detect the range of students' possession of such practices at different stages of learning. By reviewing these sources, eight practices can be identified. The first is "asking questions (for science) and defining problems (for engineering)" whereby a student's mind and thinking are triggered through brainstorming to ask questions that lead to identifying a problem. The second practice is "developing and using models" in which a student forms a mental or practical conception that explains the phenomenon or problem at hand; practices physics by constructing a model that

helps in interpretation and prediction; and then uses that model to analyze the surrounding systems to reach possible solutions. Third, in "planning and investigating," a student is placed in a position that enables him/her to practice observation, to analyze, and to conduct surveys in order to propose and test a hypothesis. The fourth practice is "analyzing and interpreting data" where a student is instructed to analyze data and information collected from previous practices using diagrams, statistical analysis, data processing and interpretation. Fifth, in "involvement with proofs and evidence," a student extracts proofs and evidence to identify points of weakness and strength in order to choose the best interpretation for investigated phenomena as well as to critique and evaluate other opinions. The sixth practice is "interpretations and solutions design" in which a student constructs theories to interpret such phenomena by using systematic solutions for solving problems and choosing optimal designs. Seventh, "obtaining, evaluating and communicating information" enables a student to read scientific texts and explains them with an aim of developing, interpreting, and evaluating information sources to verify their validity. In the eighth practice, "using mathematics and computational thinking," a student uses various mathematical and engineering skills since math and computational thinking constitute a significant part of science and engineering. In this practice, scientific models that illustrate phenomena are presented arithmetically or symbolically to provide a scientific and logical interpretation for different patterns.

Lee, Quinn & Valdes [21] pointed out that modern scientific and engineering practices are an attempt at reshaping science teaching to help students act and think like scientists and engineers. In this approach to science teaching, students are trained in forming hypotheses and testing these hypotheses by experimenting, collecting and analyzing data, forming evidence-based conclusions, formulating arguments based on findings, and ultimately designing solutions. In the scientific community, such an active, discovery-based approach to science is well-received [22]. The scientific and engineering practices were meant to be an educational departure from the sequential presentation of scientific topics in textbooks to lessons that are designed to address phenomena and from testing theoretical hypotheses to model-guided investigations. As a result, teachers need to abandon the lecturing of science through textbooks in favor of knowledge-building practices that foster the forming of arguments and conclusions [23].

Pursuant to the curricula reform movement in general and the NGSS in particular, constant comprehensive updating and developing of the teaching-education system became necessary [24], specifically the development of physics teachers performance [2]. Such a development provides teachers with the necessary competences to cope with these standards. Because previous studies [2–4, 10, 21, 24] did not directly and specifically tackle such practices of physics teachers, this study attempts to unveil the extent to which the NGSS-aligned scientific and engineering practices are present in the performance of physics teachers of the secondary stage.

## Study problem

Educators and those involved in the NGSS believe that these standards in general and scientific and engineering practices in particular should be present in the performance of physics teachers. [4, 7, 13, 18, 20, 25–31]. Therefore, the more practice they have in such standards, the better their teaching performance will be, and that will consequently be reflected in the performance of their students. However, despite the latest reforms related to involving students in physics through development of curriculum and educational environment and teacher training, there are still big challenges that confront teachers who attempt to constantly apply science practices in class [32]. There is evidence that many science classes still do not

engage students in scientific investigation and explanation [33, 34]. Therefore, the NGSS approach to science teaching requires updated training opportunities for teachers in which their scientific and engineering practices are developed and improved [23]. Such improvement will consequently reflect on students' practices and performance in science [35].

The current study, therefore, investigates the degree to which scientific and engineering practices aligned with the NGSS are present in the performance of physics teachers by answering the following two questions:

1. What is the degree to which scientific and engineering practices aligned with the NGSS are present in the performance of physics teachers at the secondary stage?

2. Do qualification and years of experience affect the degree to which the NGSS-aligned practices are present in the performance of physics teachers at the secondary stage?

## Terms of the study

**NGSS**: These are American standards that provide a new vision for teaching science in America. They comprise three integrated dimensions: main ideas, scientific and engineering practices, and shared concepts [10]. Procedurally, they are a set of standards associated with the process of teaching and learning physics at the secondary stage. They determine the standards which the students of physics should acquire, focusing on the aforementioned three dimensions. This study is focused on the scientific and engineering practices needed for teachers of physics at the secondary school stage.

**Scientific and Engineering Practices**: They are the second applicable dimension of the NGSS which incorporate eight practices related to science and engineering. These are "asking questions and defining problems"; "developing and using models"; "planning and carrying out investigations"; "analyzing and interpreting data"; "using mathematics and computational thinking"; "constructing interpretations and designing solutions"; "involvement with proofs and evidences"; and "obtaining, evaluating, and communicating information" [17, 36, 37]. The study adopted these eight practices and 44 indicators that teachers of physics at the secondary stage should possess. These practices were measured through the responses of the study subjects to the study's two instruments which were prepared for this purpose.

## Methodology

### Study approach and participants

The study adopted the mixed-methods approach (Sequential Explanatory). It employed quantitative and qualitative methods of data collection (interview and questionnaire) to get a greater understanding of the research problem. Rather than only using either quantitative or qualitative tools, "mixed methods research is a good design to use if you seek to build on the strengths of both quantitative and qualitative data" [38]. In this respect, Cohen et al. [39] argued that mixed methods "is a way of thinking, in which researchers have to see the world as integrated and in which they have to approach research from a standpoint of integrated purposes and research questions". Furthermore, conducting mixed methods studies allows researchers to tackle more complex issues and gather richer information about their topic of investigation than in studies involving only a single instrument [40].

The quantitative data of this study were collected from (49) secondary stage physics teachers in Taif city who were randomly selected for the fall semester of the 2021/2022 academic year. The schools at which these teachers worked were also chosen using simple random

selection and numbered 26 schools. The qualitative data were collected from (6) physics-teaching supervisors who were chosen purposively. Confidentiality and independence were made clear to the participants from the onset of collecting the data. All participants, teachers and supervisors, consented to participate in the study. Their consent was formally recorded using a form and a participant information sheet was given to them.

## Study instruments

To achieve its objectives, the study used the following two instruments:

- **Closed-Ended Questionnaire**

After determining the objectives of the questionnaire, the scientific and engineering practices for physics teachers at the secondary stage were identified from the official website of the NGSS (https://www.nextgenescience.org). Modern education literature such as [2–4, 7, 18, 20, 28, 30, 31, 41] was reviewed, and a number of specialists in teaching science were consulted.

The questionnaire comprised of 44 indicators representing eight practices, (S1 File of questionnaire). Responses were measured in accordance to Likert's five-point scale (very high, high, medium, weak, very weak). The questionnaire was verified for validity and reliability by applying it to an exploratory sample of (15) physics teachers. To verify the questionnaire's validity, the internal consistency (correlation coefficient) between the score of each item and the total score of the practice to which it belongs was calculated at (0.42–0.73). Next, the correlation coefficient between the total score of each practice and the total score of the entire questionnaire was calculated at (0.49–0.87). The Construct validity of the questionnaire was verified by conducting a confirmatory factor analysis (CFA) based on the Principle Components Analysis (PCA) method for the scores of the subjects on the questionnaire items, and the components were rotated using the Varimax method. The results of the factor analysis showed that there were (8) factors; the items were distributed over the eight factors, where the saturation values on the factors (factor loading) were greater than 0.3, which confirms the factor validity of the questionnaire. To verify the questionnaire's reliability, the Cronbach's alpha was calculated to be (0.88). By calculating the range (5–1 = 4) and dividing it by the largest value in the scale; to get the cell length (4÷5 = 0.80), then this value is added to the lowest value in the scale, the questionnaire responses were judged according to the following criterion: if the mean is (1.80 or less), the degree of presence is "very weak"; (more than 1.80–2.60) "weak"; (more than 2.60–3.40) "medium"; (more than 3.40–4.20) "high"; and (more than 4.20) "very high."

## Interview

Eight open-ended questions were prepared after consulting the official website of the NGSS and various studies such as [2–4, 15, 17–20], (S2 File of Interview questions). These questions were designed to elicit the viewpoint of the education supervisors regarding the degree to which scientific and engineering practices were present in the performance of physics teachers at the secondary stage. To verify the validity of the interview, the questions were presented to specialists in science teaching for their opinion.

After the supervisors were selected, a detailed description was given to them regarding the study sample, the mechanism of choice, and the procedures of preparing and conducting the interview. The supervisor's interview lasted 35 minutes. All interviews were digitally recorded and transcribed immediately after each interview. The participating supervisors were encouraged to give accurate answers, and their responses were analyzed and interpreted in depth. To

verify the reliability of the interview, the interpretation of each interview was presented to the interviewed supervisor to ensure that his point of view was correctly understood.

The analysis of the supervisors' responses was a twofold. The first part was to record their opinions as to the degree to which the scientific and engineering practices aligned with NGSS were present in the performance of the secondary stage physics teachers. The second part of the analysis concerned the reasons that the supervisors gave for their opinions. Thematic data analysis, as described in Braun & Clarke [42] was applied to the reasons given by the supervisors. In this thematic analysis, common themes for the reasons were identified, and these themes were then turned into categories.

## Study results

### The results of the first question

"What is the degree to which scientific and engineering practices aligned with the NGSS are present in the performance of physics teachers at the secondary stage?"

In answering this quest\ion, the means and standard deviations of the teachers' responses to the questionnaire and the analysis of the supervisors' interviews were calculated. The results of the questionnaire are presented in Table 1 below.

Table 1 shows that the general mean of the degree to which science and engineering practices are present in the performance of physics teachers was "medium" at (2.69). Five practices ranked "medium"; these were "asking questions and defining problems" which came first, followed by "obtaining, evaluating and communicating information"; "Planning and carrying out investigations"; "analyzing and interpreting data"; and in the fifth place "involvement with proofs and evidence". Three practices ranked "weak." These were "interpretation construction and solution design" which was sixth; "developing and using models"; and finally "using mathematics and computational thinking.", This result is corroborated by the findings of [2–4, 18, 30].

The teachers' responses to each scientific and engineering practice are detailed below.

• "Asking Questions and Defining Problems"

Table 2 shows that the general mean was (3.24) indicating a "medium" degree of presence. Three indicators ranked "high." They were (no.5) which was first, (no.4), and (no.1) in third while two indicators ranked "medium": (no.2) came fourth followed by (no.3) in fifth.

• "Developing and Using Models"

**Table 1. The means and standard deviations for teachers' responses to the scientific and engineering practices questionnaire.**

| Scientific and Engineering Practices | Mean | Standard Deviation | Degree of Presence | Rank |
|---|---|---|---|---|
| Asking questions and defining problems | 3.24 | 0.40 | Medium | 1 |
| Developing and using models | 2.51 | 0.60 | Weak | 7 |
| Planning and carrying out investigations | 3.03 | 0.65 | Medium | 3 |
| Analyzing and interpreting data | 2.74 | 0.51 | Medium | 4 |
| Using mathematics and computational thinking | 2.29 | 0.66 | Weak | 8 |
| Constructing interpretations and solution designs | 2.52 | 0.51 | Weak | 6 |
| Involvement with proofs and evidence. | 2.72 | 0.68 | Medium | 5 |
| Obtaining, evaluating, and communicating information | 3.15 | 0.33 | Medium | 2 |
| **General Mean** | **2.69** | **0.41** | **Medium** | |

**Table 2. The means and deviations of the presence of the "asking questions and defining problems" practice.**

| No. | Indicators | Mean | Standard Deviation | Degree of Presence | Rank |
|---|---|---|---|---|---|
| 1- | I ask questions to explain the factors that created a certain physical phenomenon. | 3.43 | 0.0.91 | High | 3 |
| 2- | I request students to create surveys and questions related to various physical phenomena. | 2.98 | 0.92 | Medium | 4 |
| 3- | I specify the criteria and restrictions of designing physical problems to satisfactorily reach an accurate solution for these problems. | 2.80 | 0.89 | Medium | 5 |
| 4- | I encourage students to detect and identify physical problems. | 3.49 | 0.84 | High | 2 |
| 5- | I ask questions on the data reached to identify the factors influencing them. | 3.53 | 0.82 | High | 1 |
| | **General mean** | **3.24** | **0.40** | **Medium** | |

Table 3 shows that the general mean for the presence of this practice ranked "Weak" at (2.51) with the values of the mean ranging from (2.35–2.63). Item (no.5) and (no.6) ranked "medium" with item (no.5) being first and (no.6) being second. The other remaining four indicators ranked "weak."

This result is in keeping with the findings of [2, 3, 18] in which practicing and developing models ranked "low." However, this result differs from the findings of [14, 27–30] which revealed a "good" level of the NGSS practices in teachers' performance.

- "Planning and Carrying out Investigations"

Table 4 shows that the general mean of this practice ranked "medium" at (3.03). This result mirror the findings of [2, 3, 14, 28–30] which revealed a "good" or "medium" level of the NGSS practices in teachers' performance.

- "Analyzing and Interpreting Data"

Table 5 shows that the general mean for the presence of this practice ranked "medium" at (2.74). All indicators ranked "medium" except for (no. 2) which ranked "weak" coming last. this result echoes those of [2, 3, 18, 41] which pointed out that teachers' performance in this practice ranked medium.

- Using Mathematics and Computational Thinking

Table 6 shows that the general mean for the presence of this practice ranked "weak" at (2.29) with the values of the mean ranging between (2.09–2.49) which means that all indicators ranked "weak.", This result disagrees with the findings of [14, 27–30], which revealed a

**Table 3. The means and standard deviations of the presence of the "developing and using models" practice.**

| No. | Indicators | Mean | Standard Deviation | Degree of Presence | Rank |
|---|---|---|---|---|---|
| 1- | I encourage students to develop ideas on various physical phenomena. | 2.43 | 1.00 | Weak | 5 |
| 2- | I request them to test hypotheses using the best mathematical and computational methods and to make decisions on what is or is not included in the physics model | 2.49 | 1.17 | Weak | 4 |
| 3- | I encourage them to develop causal interpretations for physical phenomenon of the real world. | 2.35 | 1.16 | Weak | 6 |
| 4- | I develop models to describe various physical phenomena. | 2.55 | 1.21 | Weak | 7 |
| 5- | I request students to explain physical phenomena using physical and mathematical terms. | 2.63 | 1.17 | Medium | 3 |
| 6- | I assess and review the teaching process via physical and mathematical acts. | 2.61 | 1.02 | Medium | 1 |
| | **General mean** | **2.51** | **0.60** | **Weak** | **2** |

**Table 4. The means and standard deviations of the presence of the "planning and carrying out investigations" practice.**

| No. | Indicators | Mean | Standard Deviation | Degree of Presence | Rank |
|---|---|---|---|---|---|
| 1- | I request students to provide evidence which shows that change in physical phenomena depends on certain forces | 3.08 | 1.03 | Medium | 2 |
| 2- | I request them to check relations between components and causes of several physical phenomena | 2.90 | 1.17 | Medium | 4 |
| 3- | I evaluate the experimental design to prove that there are certain factors which affect any physical phenomenon | 2.98 | 0.94 | Medium | 3 |
| 4- | I request students to collect data to provide evidence on the mechanism of physical phenomena formation | 3.16 | 0.65 | Medium | 1 |
| | **General mean** | **3.03** | **0.65** | **Medium** | |

satisfactory level in the performance of physics teachers in the use of mathematics and computational thinking. However, there is an agreement with [2, 3, 18] which indicated that this practice ranked "low".

- "Constructing Interpretations and Solution Design"

Table 7 shows that the general mean for the presence of this practice ranked "weak" at (2.52) with the values of the mean ranging from (2.02–2.79). Three indicators ranked "medium" with (no.7) coming first followed by (no. 5) in second and (no. 3) in third.

These results differs from [14, 29, 30] whose results regarding this practiced ranked "medium". On the other hand, it is similar to [2, 3, 18], which showed a medium and low degrees of science teachers' practice of constructing interpretations and designing solutions.

- "Involvement with Proofs and Evidence"

Table 8 shows that the general mean for the presence of this practice ranked "medium" at (2.72) with the values of the mean ranging from (2.59–2.84); (14) indicators ranked "medium" except for (no. 2) which ranked "weak.", This is in accordance with [2–4, 14, 27, 29, 30] which revealed that this practice ranked "medium" in teachers' performance.

- "Obtaining, Evaluating, and Communicating Information"

Table 9 shows that the general mean for the presence of this practice was "medium" at (3.15) with the values of the mean ranging from (2.45–3.71). Item (no.2) came first ranking

**Table 5. The means and standard deviations of the presence of the "analyzing and interpreting data" practice.**

| No. | Indicators | Mean | Standard Deviation | Degree of Presence | Rank |
|---|---|---|---|---|---|
| 1- | I request students to analyze and to interpret data to determine features of a certain physical phenomenon. | 2.92 | 1.19 | Medium | 3 |
| 2- | I request them to analyze exam data to identify similarities and differences between a number of solution designs to choose the best. | 1.90 | 1.04 | Weak | 7 |
| 3- | I request them to draw and interpret diagrams to describe certain physical correlations, | 2.96 | 1.21 | Medium | 2 |
| 4- | I encourage them to detect patterns and relations that permit using data to support a model or to interpret a result. | 2.96 | 1.21 | Medium | 6 |
| 5- | I request them to analyze and interpret data to reach a result, provide evidence, or correct scientific solution. | 2.77 | 1.21 | Medium | 5 |
| 6- | I request them to present physical data in tables. | 3.06 | 1.03 | Medium | 1 |
| 7- | I request them to analyze data statistically. | 2.90 | 1.18 | Medium | 4 |
| | **General mean** | **2.74** | **0.51** | **Medium** | |

**Table 6. The means and standard deviations of the presence of the "using mathematics and computational thinking" practice.**

| No. | Indicators | Mean | Standard Deviation | Degree of Presence | Rank |
|-----|-----------|------|--------------------|--------------------|------|
| 1- | I request students to practice mathematical thinking in analyzing physical phenomena. | 2.26 | 1.27 | Weak | 4 |
| 2- | I request them to practice computational thinking in analyzing physical phenomena. | 2.09 | 1.19 | Weak | 5 |
| 3- | I encourage them to use mathematical examples to describe simple physical models. | 2.32 | 1.28 | Weak | 2 |
| 4- | I encourage them to complete the process of data collection and to analyze large amounts of them. | 2.27 | 1.30 | Weak | 3 |
| 5- | I request them to use mathematical examples to prove how certain factors influence the increase and decrease of certain physical features or phenomenon. | 2.49 | 1.18 | Weak | 1 |
| | **General mean** | **2.29** | **0.66** | **Weak** | |

"very high.". Three indicators ranked "medium" with (no. 5) coming second, (no. 4) in third, and (no. 1) in fourth while (no. 3) came last as "weak". This indicates that the teachers did not seem to realize that gaining and evaluating information were two essential parts of scientific work.

This results parallels those of [2, 3, 18, 41] which pointed out that teachers' performance in this practice ranked "medium." However, this result diverges from [14, 27, 29, 30] which revealed the existence of a suitable level in teachers' performance regarding this practice.

## The results of the second question

"Do qualification and years of experience affect the degree to which the NGSS-aligned practices are present in the performance of physics teachers at the secondary stage?"

**Qualification.** The (t) Test was conducted to detect the significance of the differences between the means of teachers' results pertaining to the qualification variable as shown in Table 10 below.

Table 10 reveals that there were significant differences between the two means in favor of graduate teachers whose higher studies tackled modern strategies of teaching as well as international and national standards including the NGSS, and that was well-reflected in their performance. Programs of higher studies at universities emphasize modern trends in teaching, the most significant of which is currently the NGSS. The above result agrees with that of [4].

## Years of experiences

The ANOVA test was used to analyze the significance of the differences between the means of the teachers' results based on the years of experience variable (less than 5 years, from 5 years to

**Table 7. The means and standard deviations of the presence of the "constructing interpretations and solution design" practice.**

| No. | Indicators | Mean | Standard Deviation | Degree of Presence | Rank |
|-----|-----------|------|--------------------|--------------------|------|
| 1- | I request students to adopt authentic and correct evidence to explain certain physical phenomena. | 2.59 | 1.34 | Weak | 4 |
| 2- | I request them to design various physical projects. | 2.02 | 0.77 | Weak | 7 |
| 3- | I request them to apply scientific law. | 2.71 | 1.17 | Medium | 3 |
| 4- | I request them to apply ideas of physics to construct interpretations for similarities and differences between things or different phenomena. | 2.57 | 1.34 | Weak | 5 |
| 5- | I request them to construct scientific interpretations based on evidence to show how certain external factors affect a physical phenomenon | 2.73 | 1.32 | Medium | 2 |
| 6- | I request them to apply principles and scientific theories to design a way to reduce the influence of certain factors on various physical phenomena. | 2.20 | 0.96 | Weak | 6 |
| 7- | I request students to construct a scientific interpretation based on evidence that describes how differences in certain features increase the possibility of survival of the physical phenomenon. | 2.79 | 1.22 | Medium | 1 |
| | **General mean** | **2.52** | **0.51** | **Weak** | |

**Table 8. The means and standard deviations of the presence of the "involvement with proofs and evidence" practice.**

| No. | Indicators | Mean | Standard Deviation | Degree of Presence | Rank |
|---|---|---|---|---|---|
| 1- | I encourage students to present assessments and justifications for constructing physical models and scientific explanations. | 2.83 | 1.05 | Medium | 2 |
| 2- | I request them to use the claims/hypotheses supported by evidence and experiment to clarify different physical systems. | 2.59 | 1.19 | Weak | 5 |
| 3- | I request them to compare different models and interpretations that show points of strength and weakness. | 2.63 | 1.25 | Medium | 4 |
| 4- | I request them to use verbal and written proofs to support or refute a model or an interpretation of a certain physical phenomenon. | 2.69 | 1.06 | Medium | 3 |
| 5- | I encourage them to explain evaluations and evidence for each other. | 2.84 | 1.12 | Medium | 1 |
| | General mean | 2.72 | 0.68 | Medium | |

**Table 9. The means and standard deviations of the presence of the "obtaining, evaluating, and communicating information" practice.**

| No. | Indicators | Mean | Standard Deviation | Degree of Presence | Rank |
|---|---|---|---|---|---|
| 1- | I request students to distinguish between observation and induction, request and evidence, proof and interpretation. | 3.14 | 0.74 | Medium | 4 |
| 2- | I request them to use diverse means of communication patterns such as diagrams, models, and equations. | 3.71 | 0.76 | High | 1 |
| 3- | I request them to read scientific physics texts and interpret them | 2.45 | 1.32 | Weak | 5 |
| 4- | I request them to produce a scientific text in order to develop and explain a physics model. | 3.20 | 0.71 | Medium | 3 |
| 5- | I request students to collect data in order to describe a certain physical phenomenon. | 3.24 | 0.75 | Medium | 2 |
| | General mean | 3.15 | 0.33 | Medium | |

**Table 10. The (t) test results to detect statistical differences between the means of physics teachers' responses pertaining to the qualification variable.**

| Qualification | N | mean | Std. Deviation | T | df | Sig. |
|---|---|---|---|---|---|---|
| Bachelor's | 31 | 2.65 | 0.23 | 3.59 | 47 | 0.001 |
| Graduate | 18 | 2.88 | 0.18 | | | |

**Table 11. The ANOVA test results for analyzing the means of the physics teachers' responses to the years of experience variable.**

| Variance Source | Sum of Squares | df | Mean Square | f | Sig. |
|---|---|---|---|---|---|
| Between Groups | 0.22 | 2 | 0.11 | 1.97 | 0.151 |
| Within Groups | 2.593 | 46 | 0.056 | | |
| Total | 2.815 | 48 | | | |

10 years, more than 10 years) as presented in Table 11 below. The ANOVA test was chosen because the study's sample meets the requirements of parametric statistics and because the years of experience variable is of three levels.

Table 11 shows that there were no statistically-significant differences between the two means. This lack of significance can be attributed to the fact that all teachers regardless of their years of experience had been recently introduced to the NGSS. The lack of in-service training in the NGSS practices can be another factor. The above result is in accordance with [2, 4] which revealed that there were no differences with statistical significance related to the years of experience variable.

## Discussion

This study sought to unravel the degree to which scientific and engineering practices complied with the NGSS (Next Generation Science Standards) were present in the performance of physics teachers at the secondary stage in Taif city, Saudi Arabia. The study also sought to identify the effect of the qualification and years of experience variables on the degree to which the NGSS-aligned practices were present in the performance of physics teachers. This study thereby contributed to revealing the need for physics teachers to reconsider their teaching practices to align with the NGSS

The results shows the physics teachers possessed a limited awareness of the significance of these practices, and this limitedness was reflected in their performance. This can be attributed to some factors such as the shortcomings of some components in teachers-training programs during or before service; teaching and administrative overload; and the inadequacies of lab equipment in many schools. According to Gurt et al [43], teachers need to experience scientific and engineering practices as learners first to be able to teach them in the classroom. It is necessary, therefore, for physics teachers to be provided with pre-service and during-service training programs that target the development of their scientific and engineering practices so that they are able to engage their students in these practices as prescribed in the NGSS [44, 45].

• "Asking Questions and Defining Problems"

These results revealed that the physics teachers were somewhat keen on asking questions to check the range of the students' comprehension of the data and making sure the students connected what they learned to the physical phenomena and problems they encountered in life. The table also shows that the physics teachers tackled physical problems taken from the students' life by asking them to identify these problems, feel them, and define them in order to offer suitable solutions. This result agrees with the findings of the following studies: [2–4, 14, 27–30, 41].

The results of the supervisors' interviews revealed that the supervisors believed the physics teachers asked questions in an acceptable way. The supervisors attributed this result to the nature of teaching physics which is centered on training students to ask questions since physics presents students with many perplexing phenomena that require careful investigation through asking questions. This nature prompted the teachers to ask questions during class and present scientific problems so that the students could ask their own questions about them.

All supervisors agreed that the teachers' performance was lower than what was expected when it came to formulating scientific problems for students, and that negatively affected the student's skill in defining scientific problems. The supervisors claimed that this shortcoming was due to the inadequate training of the teachers in this domain since teacher-training programs were mostly theoretical. The supervisors' responses were in accordance with the findings of [2, 3, 28, 41].

• "Developing and Using Models"

Although physics necessitates using scientific predictions for various phenomena, which obligates using models, the findings revealed that many teachers still did not possess this practice. The absence of this practice can be explained by the failure of teacher-training programs to help teachers acquire the skill of model building. In addition, heavy teaching loads, time constraints, and long school curricula can be seen as contributing factors.

The analysis of the supervisors' interviews revealed an agreement between their viewpoints and the teachers'. Two supervisors pointed out that the physics teachers were clearly weak with regard to developing and using models in teaching; most of them still used traditional methods

of teaching that depended on lecturing. They also revealed that the long science curriculum, the heavy teaching loads, and administrative work obstructed the teachers from using and developing models. Despite these challenges, the supervisors noted that the physics teachers understood how their students think. Accordingly, they asked the students to correlate relations between phenomena and diversified systems and ultimately evaluate and revise them.

Two other supervisors pointed out that some teachers showed an improvement in using modern teaching strategies in general and models in particular as a result of being graduate students who study courses that address these strategies.

- "Planning and Carrying out Investigations"

Although the nature of the subject matter of physics requires experimentation and scientific report writing, the teachers revealed that their practice of "planning and carrying out investigations" was "medium." This inadequacy indicates that the practical application aspect of teaching physics was still below the required level. It also shows that teacher-training programs need to focus more on the practical application side of physics and to highlight the importance of scientific experiments in teaching physics. In addition, it is important for students to acquire the skills of data collecting via sound scientific methods and to use this data correctly in explaining scientific phenomena. [7] believe that teachers should ask their students to conduct scientific investigations and provide proofs and evidence for the results they reach.

The results of the interviews revealed that all the supervisors believed that while teachers were convinced of the importance of planning in teaching sciences, the majority still used ready plans. The supervisors, thus, noticed a difference between what is written in the daily or semester plans and the actual performance of the teacher in the classroom. This throws in doubt the credibility of the planning records that teachers had which complicates evaluating teachers in the skill of lesson planning. One way to remedy this situation, the supervisors think, is to create programs and incentives that encourage teachers to practice good planning.

The supervisors added that many teachers did not understand the concept of surveying; consequently, they used the traditional method of teaching. As a result, the students' level of surveying skills was generally weak.

- "Analyzing and Interpreting Data"

This result shows that some teachers still did not realize the significance of students possessing the practice of analyzing and interpreting scientific data which is essential in solving problems that students encounter in their real life. Therefore, teachers need to be made aware of its significance through teacher-training programs. Without scientific analysis, one cannot accurately identify and comprehend correlations, causes, system components, and scientific phenomena. Moreover, understanding many topics of physics requires comparisons between such topics and other phenomena, this entails analyzing every topic or phenomenon separately then make the required comparison. Therefore, it is essential that teachers have the desire and motivation to familiarize their students with the importance of reliable scientific evidence, and one of the best ways of getting such evidence is analyzing the data collected from different scientific phenomena.

Analyzing the responses to the interview question about the data analysis and interpretation practice, all supervisors confirmed that the physics teachers practiced this skill but not at the required level. The supervisors attributed this weakness to a number of reasons: the traditional teaching method teachers still used; the lack of data-analysis components in teacher-training programs; the teachers reliance on ready-made teaching plans and data analyses; the large number of students in classes; and the teaching and administrative overloads.

- "Using Mathematics and Computational Thinking"

The results shows that the presence of this practice ranked "weak", this result can be attributed to the nature of physics where students rely heavily on math and diagrams to understand and solve problems. In addition, many teachers still had poor skills in model building which hindered them in relating mathematical examples to scientific models. Moreover, the insufficient technical teaching aids at schools contributed to the low level of practicing computational thinking.

Analyzing the results of the supervisors' interviews, it was noted that they all confirmed that the physics teachers practiced mathematical thinking during the teaching process but to a limited extent. Few teachers assigned their students scientific tasks whose solutions needed math and mathematical thinking. As for computational thinking, the supervisors also pointed out that its use and practice were lacking as well. They attributed that to several reasons: the teacher-training programs deficiency in this practice; the low level of teachers awareness; the shortage of equipment and computers at some schools; the teaching and administrative overloads; and the large number of students in classes. They indicated that solving such a problem was important and could be achieved through providing teachers who do this practice with moral and material incentives.

- "Constructing Interpretations and Solution Design"

The results reflected that teachers were still below the expected level in their conviction of the importance of students' need to possess scientific evidence to present their ideas or explain physical phenomena. The rest of indicators ranked "weak.", Although physics requires students to make scientific comparisons based on evidence and applying knowledge and scientific laws, the physics teachers still lacked the practices needed to achieve such a goal because of the dependence on the traditional methods of teaching.

Analyzing the supervisors' interviews revealed that the viewpoints of two of them agreed with the questionnaire results: that not only teachers practiced data interpretation and solution design to a great extent in teaching, they also assigned students tasks to design solutions for scientific problems. The other five supervisors, however, pointed out that this practice was not that much followed. They attributed this to the insufficient class time which does not help the teacher to do this practice in addition to the shortage of teachers who are proficient in this practice. The supervisors added that the clearly-stated solution designs found in teachers' guides were often overlooked by the teachers. Thus, teaching physics remained traditional depending solely on memorization. This opinion agrees with the findings of [2, 3].

- "Involvement with Proofs and Evidence"

It can be noticed from the results that the teachers performance in this practice was lacking; therefore, some of them avoided asking students to evaluate or explain any phenomenon. Heavy teaching and administrative loads as well as the large number of students in classes also complicated the teachers' use of this practice. As a result, teachers are still in need of training and rehabilitation in constructing models and teaching it to their students.

The results of the supervisors' interviews agree, to an extent, with those of the questionnaire which revealed an acceptable degree of this practice that was manifested in providing evidence while presenting scientific knowledge, even via the traditional method, though most of that is done theoretically. The practical and applicable proofs were rarely used because of the lack of suitable equipment and labs at schools in addition to short class time, the large number of students, and the teaching and administration burdens.

- "Obtaining, Evaluating, and Communicating Information"

The results shows that the presence of this practice was "medium", this result can be attributed to the nature of physics which necessitates using diagrams and scientific laws. and this indicates that the teachers did not seem to realize that gaining and evaluating information were two essential parts of scientific work.

Therefore, teachers of physics need to be made aware of the importance of students identifying and using suitable and authentic sources of data and information in order to benefit from them in learning and in real life. Teachers also need to place more emphasis on scientific reading and encourage students to practice scientific reading silently and loudly. In addition, students should be instructed to concentrate on all basic scientific and integrated processes and be provided with educational situations that enable them to practice each processes clearly and independently.

The analysis of the supervisors' interviews revealed a consensus that a number of teachers never deviated from the textbook as the source of information. They attributed this behavior to administrative and technical reasons associated with time constraints, the lack of good training, and the shortage in technical labs at schools. In addition, some teachers, the supervisors believe, did not know how to deal with the sources of technical information. However, they pointed out that the teachers requested students to read scientific texts, but they rarely asked them to invent or produce scientific texts for the sake of designing scientific models or explaining certain phenomena. The supervisors' opinions are in keeping with the findings of [2, 3, 18, 41].

- Differences according to qualification and years of experience

The results showed that graduate programs in universities focus on recent trends in teaching, especially on NGSS standards which contributed to better application of science and engineering practices by teachers with graduate degrees than teachers with Bachelor's degree. The results also showed that teachers' years of experience did not affect their possession of these practices, because the NGSS standards are new for physics teachers, and this requires increasing their knowledge of these standards along with the scientific and engineering practices.

## Study limits and limitations

This study was limited to a dimension (scientific and engineering practices), as one of the main dimensions of NGSS, Because it is related to the teacher's teaching performance, The study was also confined to measuring scientific and engineering practices contained in the study's instrument (the questionnaire) as one of the three major dimensions of the NGSS. It was also limited to a sample of physics teachers at the secondary stage in Taif city for the fall semester of the 2021/2022 academic year, in addition to only six education supervisors who agreed to participate. The study relied on the opinions of the participants, so there is a certain level of subjectivity. Furthermore, being conducted in only one city, Taif, makes it unknown whether the results of this study can be generalized to other teachers of other cities. However, this study was conducted in public secondary schools comparable to many others across Saudi Arabia, and it took into consideration the viewpoints of as many physics teachers as possible. Results were determined by the range of validity and reliability of the study's two instruments (the questionnaire and the interview).

## Conclusion

As stated in the introduction, to implement the NGSS, there is an urgent need to conduct studies on the scientific and engineering practices in teaching physics [8]. The purpose of this study is to detect the degree to which eight NGSS-aligned engineering and scientific practices

are present in the performance of physics teachers at the secondary stage, from their own perspective and the perspective of education supervisors.

The results revealed that the viewpoints of the two groups, the teachers and supervisors, were concordant to a great extent. teachers did not fully comprehend all the scientific and engineering practices which were expected to be mastered by the students. Therefore, these practices were not manifested as required in their performance. Consequently, the teachers need sufficient support to enable them to completely understand these practices in a way that complies with the NGSS so as to eventually be able to help their students acquire them. This support can be achieved when the Ministry of Education takes into consideration these practices and train teachers in them. The courses offered by colleges of education also need to be reshaped in a way that prepares physics teachers in light of the NGSS. Finally, there is also a need to practically apply these practices in the teacher-training programs before and during service. It is recommended that similar studies are applied to the teachers of the primary and elementary stages. It is also recommended that experimental studies be conducted on training programs that prepare teachers in the NGSS-aligned scientific and engineering practices.

## Supporting information

**S1 File Study instrument questionnaire.**
(PDF)

**S2 File Study instrument interview.**
(PDF)

## Acknowledgments

The author would like to express his thanks to the physics teachers and education supervisors involved in this study for their time and engagement

## Author Contributions

**Conceptualization:** Mohammad Khair M. Al-Salamat.

**Data curation:** Mohammad Khair M. Al-Salamat.

**Formal analysis:** Mohammad Khair M. Al-Salamat.

**Funding acquisition:** Mohammad Khair M. Al-Salamat.

**Investigation:** Mohammad Khair M. Al-Salamat.

**Methodology:** Mohammad Khair M. Al-Salamat.

**Project administration:** Mohammad Khair M. Al-Salamat.

**Resources:** Mohammad Khair M. Al-Salamat.

**Software:** Mohammad Khair M. Al-Salamat.

**Supervision:** Mohammad Khair M. Al-Salamat.

**Validation:** Mohammad Khair M. Al-Salamat.

**Visualization:** Mohammad Khair M. Al-Salamat.

**Writing – review & editing:** Mohammad Khair M. Al-Salamat.

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
