## [Decision Letter · Decision Letter 0]

6 Jan 2022

PONE-D-21-34137Scientific and Engineering Practices Aligned with NGSS Standards in the Performance of Secondary Stage Physics TeachersPLOS ONE

Dear Dr. Alsalamat,

Thank you for submitting your manuscript to PLOS ONE. After careful consideration, we feel that it has merit but does not fully meet PLOS ONE’s publication criteria as it currently stands. Therefore, we invite you to submit a revised version of the manuscript that addresses the points raised during the review process.

We look forward to receiving your revised manuscript.

Kind regards,

Yiming Tang, Ph.D.

Academic Editor

PLOS ONE

Journal Requirements:

Reviewers' comments:

Reviewer's Responses to Questions

**Comments to the Author**

1. Is the manuscript technically sound, and do the data support the conclusions?

Reviewer #1: No

Reviewer #2: Partly

Reviewer #3: Partly

2. Has the statistical analysis been performed appropriately and rigorously? 

Reviewer #1: No

Reviewer #2: Yes

Reviewer #3: No

3. Have the authors made all data underlying the findings in their manuscript fully available?

Reviewer #1: No

Reviewer #2: Yes

Reviewer #3: Yes

4. Is the manuscript presented in an intelligible fashion and written in standard English?

Reviewer #1: No

Reviewer #2: Yes

Reviewer #3: Yes

5. Review Comments to the Author

Reviewer #1: This article employed a survey methodology to examine self-reporting of adherence to Next Generation Science Standards (NGSS) by secondary education physics teachers in Saudi Arabia. While the authors demonstrate a clear interest in furthering science education by exploring the NGSS, a variety of issues prevent the reviewer from recommending this article for publication. These include:

Overview/Statement of the Problem/Significance

- The sentence "Many institutions catered for developing the field of teaching and learning

science, relying on such standards" does not have a clear meaning and should be rewritten to clarify the subject, verb, and object of the sentence.

- Early in the manuscript, the authors need to clarify that NGSS stands for "Next Generation Science Standards."

- The introductory paragraph consists mainly of the authors' assertions without a strong evidentiary base. If an overview of the standards is warranted here, it should be grounded in evidence-based claims from reputable sources.

- The citation for the Next Generation Science Standards documents does not match the information provided on the NGSS website itself regarding copyright and trademark: https://www.nextgenscience.org/trademark-and-copyright

- Assertions such as "Because students’ objectives for learning sciences in general and physics in particular have changed,

science teachers of all levels need to adapt themselves to these new standards in order to achieve an actual change in

their teaching practices. Therefore, educators see that teachers’ qualifying programs should be developed to make

teachers concentrate more on actual practice and to help them amend their practices." (p. 2) appear throughout the manuscript and lack references and citations to support them. Because of this pattern, many of the premises that follow from these assumptions appear unfounded.

- Despite claims by the authors, a goal of the NGSS "to make students behave and practice in a scholarly manner to gain skills of engineering design which enable them to conduct research and to solve problems they encounter throughout science study or in their actual life" appears nowhere in the standards documents. For reference: https://www.nextgenscience.org/faqs

- The Eight Science and Engineering practices described on page 3 are written in gender-biased language (deictically referring to "student" as "he/him/his." Further, typeface choices such as setting certain words in boldface are distracting to the reviewer.

Review of Literature

- There is no clear section that reviews literature; therefore, it is unclear how the current manuscript is building on or extending prior work in this field.

Methodology

- While the study refers to a "mixed approach" to research, the author should clarify whether this was a mixed methods design (and if so, how), or whether there were simply multiple methods or multiple sources of data.

- Consent procedures should specify whether oral or written consent was provided by participants.

- Study limitations are not discussed with sufficient detail. A much clearer explication of the sampling frame is necessary to evaluate the procedures of random and purposive selection. A check for validity was referenced, although there is no clear discussion of whether this is for construct, content, face, or criterion validity. Further, despite the mentioning of an exploratory survey, no statistical calculations or data are provided to support the authors claims of validity and reliability. In the case of qualitative trustworthiness, there is no mention whatsoever of credibility, transferability, dependability, confirmability, or authenticity.

- There are no clearly grounded definitions for the metric of "degree of availability," and thus, it appears to be an arbitrary construction of the authors. If any rating scale is to be used, it should be one that is grounded in prior evidence and tested for validity and reliability through established statistical analysis.

- There is no clear discussion of how qualitative interview data was analyzed, nor of how the conclusions were reached.

- Given the above statistical issues, the ANOVA does not appear to offer valid or reliable results.

With a much clearer discussion of the overall study design, sampling methodology, calculations of validity and reliability, qualitative trustworthiness, and evidence-based definition of the term "degree of availability," the study may be eventually become suitable for publication. However, in its current state, this reviewer believes it does not meet the standards of the journal.

Reviewer #2: 1. PLOS uses “Vancouver” style, as outlined in the ICMJE sample references. Please modify it.

2. I suggest that the author move the section "Study limits and limitations" to the paragraph "Conclusion".

3. The source of the basis for the evaluation criteria of the questionnaire explored?

4. Do these eight open-ended questions have relevant literature basis?

5. The problem of demographic variables was not seen in the research.

6. The teacher’s seniority is a continuous variable. Why the F test is used in this study, please add the author's explanation.

Reviewer #3: Abstract

1. There is no problem statement.

2. It is not stated what questionnaire was used.

3. The implications of the findings are not clearly stated in the abstract

Introduction

1. NGSS is the first word mentioned on page 1, but it is an acronym and it is not shown what it stands for.

2. The introduction and theoretical background is rather long. The format of having subtopics “significance of the study, terms of the study are from the dissertation format.

3. There are many standards mentioned such as NSTA, NSF, AAAS but what is the purpose for and how is it different from the NCSS? What is the change and adaptation that needs to be done? This needs to be more specific.

4. There were 8 practices outlined on page 3. These practices should be refenced. And literature can be added to build upon it as it seems to form the theoretical background of the study.

5. On page 4: “Educators and those concerned with NGSS see that physics teachers should possess such standards in general” Do teachers “possess standards?

6. In the research questions there is mention of degree of availability ie “What is the degree of availability pertaining scientific and engineering practices that cope with NGSS in the performance of physics teachers at the secondary stage? What exactly does this term mean?

7. Is there any hypothesis for the study.

8. On page 5, it was stated that this is a mixed method approach. Is there a refence for the framework of this mixed method study?

9. Who exactly are the participants? If they are teachers, why do they need supervisors? Or are these preservice teachers?

10. On page 6, it was written: “After determining objectives of the questionnaire, the researcher checked the official website of NGSS (https:www.nextgenescience.org), modern education literatures, besides surveying scientific and engineering practices relevant to teachers of physics at the secondary stage as presented in (Malkawi & Rababah, 2018 ; Hang & Srisawasdi, 2021 ; Alshyab, 2019 ; Aboathrah, 2019 ; Brownstein & Horvath, 2016 ; Qablan, 2016 ; Fulcher, 2014 ; Stuart et al., 2021 ; Duschl & Bybee, 2014 ; Kawasaki, 2015), and ..

a. The writing of in-text citation above is incorrect.

b. The validation of the instrument is also not clear.

11. The presentation of the findings is not well done. eg. on page 8: The item of teachers’ scientific and engineering practices might be elaborated on as follows:

•Asking questions and defining problems

Table 2. presents results pertaining this type of practice.

12. The second RQ on page 19: Does that degree of availability in performance of those teachers vary in

accordance with qualification, or years of experience?", was analysed using t-tests. However, the variables seem to be of nominal and ration data respectively. Is t-test the correct means of analysis.

13. What are the implications to the study. And the recommendations should be made more specifically.

6. PLOS authors have the option to publish the peer review history of their article (what does this mean?). If published, this will include your full peer review and any attached files.

Reviewer #1: **Yes: **Earl Aguilera

Reviewer #2: No

Reviewer #3: No

---

## [Author Response · Author response to Decision Letter 0]

11 Feb 2022

Reviewer # 1

Comment 1

Overview/Statement of the Problem/Significance

- The sentence "Many institutions catered for developing the field of teaching and learning

science, relying on such standards" does not have a clear meaning and should be rewritten to clarify the subject, verb, and object of the sentence.

Response: Modified. The paragraph was rephrased.

Comment 2: Early in the manuscript, the authors need to clarify that NGSS stands for "Next Generation Science Standards."

Response: Modified. Clarified the first time it was used.

Comment 3:The introductory paragraph consists mainly of the authors' assertions without a strong evidentiary base. If an overview of the standards is warranted here, it should be grounded in evidence-based claims from reputable sources. 

Response: Modified. A number of studies were cited to corroborate the author’s assertions.

Comment 4: The citation for the Next Generation Science Standards documents does not match the information provided on the NGSS website itself regarding copyright and trademark

Response: The theoretical background on the NGSS was expanded and supported. However, it is not clear what the reviewer means in this remark when saying “regarding copyright and trademark.” Further explanation is needed for the author to be able to address this issue.

Comment 5: - Assertions such as "Because students’ objectives for learning sciences in general and physics in particular have changed,

science teachers of all levels need to adapt themselves to these new standards in order to achieve an actual change in

their teaching practices. Therefore, educators see that teachers’ qualifying programs should be developed to make

teachers concentrate more on actual practice and to help them amend their practices." (p. 2) appear throughout the manuscript and lack references and citations to support them. Because of this pattern, many of the premises that follow from these assumptions appear unfounded.

Response: Modified. Author’s assertions have been backed up by previous research including the example provided.

Comment 6: Despite claims by the authors, a goal of the NGSS "to make students behave and practice in a scholarly manner to gain skills of engineering design which enable them to conduct research and to solve problems they encounter throughout science study or in their actual life" appears nowhere in the standards documents. For reference: https://www.nextgenscience.org/faqs

Response: Modified. This sentence was misphrased as a goal of the NGSS. It has been phrased correctly as a claim by previous research which has been cited. 

Comment 7:The Eight Science and Engineering practices described on page 3 are written in gender-biased language (deictically referring to "student" as "he/him/his." Further, typeface choices such as setting certain words in boldface are distracting to the reviewer.

Response: Modified. The practices have been rephrased bearing in mind gender-neutrality and font issues as described in the comment here.

Comment 8: There is no clear section that reviews literature; therefore, it is unclear how the current manuscript is building on or extending prior work in this field.

Response: The Introduction section has been renamed “Introduction and Literature Review” In this section, the topic of the study is introduced along with reviewing previous research on the topic. This is why no separate section for (literature review) was created. 

Comment 9: While the study refers to a "mixed approach" to research, the author should clarify whether this was a mixed methods design (and if so, how), or whether there were simply multiple methods or multiple sources of data.

Response: Modified. A description of “mixed approach” has been added to the “Study Approach and Participants” section. 

Comment 10: Consent procedures should specify whether oral or written consent was provided by participants.

Response: Modified. The consent procedures in the “Study Approach and Participants” section now explains that the consent was a written one.

Comment 11: Study limitations are not discussed with sufficient detail. 

Response: More details were added to the “Study Limitations” section. 

Comment 12: A much clearer explication of the sampling frame is necessary to evaluate the procedures of random and purposive selection. 

Response: More details about the sampling of the participants were added to the “Study Approach and Participants” section

Comment 13: A check for validity was referenced, although there is no clear discussion of whether this is for construct, content, face, or criterion validity. Further, despite the mentioning of an exploratory survey, no statistical calculations or data are provided to support the authors claims of validity and reliability. 

Response: More details were added in the “Questionnaire” and “Interview” sections to explain how validity and reliability were tested and verified.

Comment 14: In the case of qualitative trustworthiness, there is no mention whatsoever of credibility, transferability, dependability, confirmability, or authenticity.

Response: The qualitative trustworthiness has been elaborated, and the whole “Interview” section was modified

Comment 15: There are no clearly grounded definitions for the metric of "degree of availability," and thus, it appears to be an arbitrary construction of the authors. If any rating scale is to be used, it should be one that is grounded in prior evidence and tested for validity and reliability through established statistical analysis.

Response:The metric is provided in the “Questionnaire” section. It is as follows: “The questionnaire responses were judged according to the following criterion: if the mean is (1.80) or less, the degree of presence is “very weak”; (more than 1.80 -2.60) “weak”; (more than 2.60-3.40) “medium”; (more than 3.40-4.20) “high”; and (more than 4.20) “very high.”

Comment 16: There is no clear discussion of how qualitative interview data was analyzed, nor of how the conclusions were reached.

Response: Modified. A separate paragraph was added explaining how the qualitative data was analyzed.

Comment 17: Given the above statistical issues, the ANOVA does not appear to offer valid or reliable results.

Response: Since the above statistical issues have been addressed, the ANOVA should deliver reliable results. In addition, the statistical analysis of this study was presented to a specialist in educational statistics who approved of the reliability of the ANOVA test results in this study.

Reviewer # 2

Comment 1: PLOS uses “Vancouver” style, as outlined in the ICMJE sample references. Please modify it.

Response: Modified. Vancouver Style used.

Comment 2: I suggest that the author move the section "Study limits and limitations" to the paragraph "Conclusion".

Response: The placement of the “Study Limits and Limitations” section before the “Results” section is in keeping with a convention in research writing. If the reviewer is giving a choice of replacing this section in the “Conclusion,” the author would like to keep it as is. It is worth mentioning that the study’s limits and limitations were restated in the second paragraph of the “Conclusion” section.

Comment 3:The source of the basis for the evaluation criteria of the questionnaire explored?

Response: Modified. More details about the questionnaire were added in the “Questionnaire” section to address this issue. 

Comment 4: Do these eight open-ended questions have relevant literature basis?

Response: Yes. They were based on the same literature from which the eight scientific and engineering practices were derived. The “Interview” section has been modified to clarify this point. 

Comment 5: The problem of demographic variables was not seen in the research.

Response: The demographics of the participants were mentioned in the “Study Questions” section (question 2) and they were elaborated in the discussion of the results.

Comment 6: The teacher’s seniority is a continuous variable. Why the F test is used in this study, please add the author's explanation.

Response: Added. The explanation of why the F test was used in this study was added in the “Results” section when addressing the “years of experience” variable.

Reviewer # 3

Comment 1:There is no problem statement

Response: Modified. Problem statement added

Comment 2: It is not stated what questionnaire was used.

Response: The study used a closed-ended questionnaire. The type was added to the “Abstract” section and the “Questionnaire” section. 

Comment 3:The implications of the findings are not clearly stated in the abstract

Response: Modified. Implications were stated in the abstract.

Comment 4: NGSS is the first word mentioned on page 1, but it is an acronym and it is not shown what it stands for.

Response: Modified. Acronym explained

Comment 5:The introduction and theoretical background is rather long. The format of having subtopics “significance of the study, terms of the study are from the dissertation format.

Response: This section has been renamed “Introduction and Literature Review” based on one of the remarks of Reviewer # 1. Therefore, it ended up being a lengthy section because it tackles two parts, i.e. introducing the topic and reviewing the relevant literature.

Comment 6: There are many standards mentioned such as NSTA, NSF, AAAS but what is the purpose for and how is it different from the NGSS? What is the change and adaptation that needs to be done? This needs to be more specific.

Response: The NGSS build on the previous standards. They are a progression from these standards. This relationship is stated in the transitions between the first and the second paragraphs of the “Introduction” section.

Comment 7: There were 8 practices outlined on page 3. These practices should be refenced. And literature can be added to build upon it as it seems to form the theoretical background of the study.

Response: The 8 practices were clearly referenced at the beginning of the paragraph where they are presented.

More literature has been added to substantiate these practices. 

Comment 8: On page 4: “Educators and those concerned with NGSS see that physics teachers should possess such standards in general” Do teachers “possess standards?

Response: The sentence has been rephrased as the following: “Educators and those involved in the NGSS believe that these standards in general should be present in the performance of physics teachers.”

Comment 9: In the research questions there is mention of degree of availability ie “What is the degree of availability pertaining scientific and engineering practices that cope with NGSS in the performance of physics teachers at the secondary stage? What exactly does this term mean?

Response: Modified. The phrasing of the study’s title, problems, and questions has been changed.

So, the question now reads, “What is the degree to which scientific and engineering practices aligned with the NGSS are present in the performance of physics teachers at the secondary stage? 

Comment 10: Is there any hypothesis for the study.

Response: No, there isn’t.

Comment 11: On page 5, it was stated that this is a mixed method approach. Is there a reference for the framework of this mixed method study?

Response: Modified. This section was rewritten, and the mixed method was explained and referenced. 

Comment 12: Who exactly are the participants? If they are teachers, why do they need supervisors? Or are these preservice teachers?

Response: The participants are teachers at the secondary stage in addition to educational supervisors. In Saudi Arabia’s educational system, educational supervisors are in charged with monitoring the performance of teachers and providing feedback to them. 

Comment 13:On page 6, it was written: “After determining objectives of the questionnaire, the researcher checked the official website of NGSS (https:www.nextgenescience.org), modern education literatures, besides surveying scientific and engineering practices relevant to teachers of physics at the secondary stage as presented in (Malkawi & Rababah, 2018 ; Hang & Srisawasdi, 2021 ; Alshyab, 2019 ; Aboathrah, 2019 ; Brownstein & Horvath, 2016 ; Qablan, 2016 ; Fulcher, 2014 ; Stuart et al., 2021 ; Duschl & Bybee, 2014 ; Kawasaki, 2015), and ..

a. The writing of in-text citation above is incorrect.

b. The validation of the instrument is also not clear.

Response: a. Modified according to Vancouver Style, b. Modified. The validity of the instrument has been elaborated.

Comment 14: The presentation of the findings is not well done. eg. on page 8: The item of teachers’ scientific and engineering practices might be elaborated on as follows: •Asking questions and defining problems, Table 2. presents results pertaining this type of practice.

Response: It is not clear what is required here. An example is provided, but it is not explained what is wrong with it.

In this study, the eight scientific and engineering practices were formulated as the main items of the questionnaire and were assigned a number of questions. The responses to the questions pertaining to each practice were presented in a separate table and analyzed afterwards. This is considered an acceptable convention of presenting and analyzing data in research writing. 

Comment 15: The second RQ on page 19: Does that degree of availability in performance of those teachers vary in

accordance with qualification, or years of experience?", was analysed using t-tests. However, the variables seem to be of nominal and ration data respectively. Is t-test the correct means of analysis.

Response: Yes. The t-test is the correct means of analysis for this question. This test was chosen based on the consultation of a specialist in statistical analysis.

Comment 16: What are the implications to the study. And the recommendations should be made more specifically

Response: Modified. Specific recommendations have been added.

---

## [Decision Letter · Decision Letter 1]

29 Apr 2022

PONE-D-21-34137R1Scientific and engineering practices aligned with the NGSS in the performance of secondary stage physics teachersPLOS ONE

Dear Dr. Alsalamat,

Thank you for submitting your manuscript to PLOS ONE. After careful consideration, we feel that it has merit but does not fully meet PLOS ONE’s publication criteria as it currently stands. Therefore, we invite you to submit a revised version of the manuscript that addresses the points raised during the review process.

We look forward to receiving your revised manuscript.

Kind regards,

Yiming Tang, Ph.D.

Academic Editor

PLOS ONE

Reviewers' comments:

Reviewer's Responses to Questions

**Comments to the Author**

1. If the authors have adequately addressed your comments raised in a previous round of review and you feel that this manuscript is now acceptable for publication, you may indicate that here to bypass the “Comments to the Author” section, enter your conflict of interest statement in the “Confidential to Editor” section, and submit your "Accept" recommendation.

Reviewer #2: All comments have been addressed

Reviewer #3: (No Response)

Reviewer #4: All comments have been addressed

Reviewer #5: (No Response)

Reviewer #6: (No Response)

Reviewer #7: All comments have been addressed

Reviewer #8: (No Response)

2. Is the manuscript technically sound, and do the data support the conclusions?

Reviewer #2: Yes

Reviewer #3: No

Reviewer #4: Yes

Reviewer #5: (No Response)

Reviewer #6: No

Reviewer #7: Yes

Reviewer #8: Partly

3. Has the statistical analysis been performed appropriately and rigorously? 

Reviewer #2: Yes

Reviewer #3: Yes

Reviewer #4: Yes

Reviewer #5: (No Response)

Reviewer #6: No

Reviewer #7: I Don't Know

Reviewer #8: (No Response)

4. Have the authors made all data underlying the findings in their manuscript fully available?

Reviewer #2: Yes

Reviewer #3: Yes

Reviewer #4: No

Reviewer #5: No

Reviewer #6: Yes

Reviewer #7: Yes

Reviewer #8: Yes

5. Is the manuscript presented in an intelligible fashion and written in standard English?

Reviewer #2: Yes

Reviewer #3: No

Reviewer #4: Yes

Reviewer #5: Yes

Reviewer #6: No

Reviewer #7: Yes

Reviewer #8: No

6. Review Comments to the Author

Reviewer #2: The content has been amended to better reflect the work conducted.

The author has made a clear description of the source of the questionnaire.

Reviewer #3: 1. Abstract does not indicate the need for the study.

2. On page 2. The problem statement is not clear. You have mentioned that science teachers of all levels need to adapt to the new standards. However, you have not shown evidence on why this is needed. So, what is the problem? Is it a difficulty in designing appropriate training? Or do we need to have a change of attitude? This is not clear.

3. On page 3, you have stated the practices in NGSS.

a. However, it is not clear why and how you have applied these practices. The practices are the standards the students need to achieve, so is there a difference with the teaching standards or the standards for teachers?

b. Similarly, in the problem statement on page 4, it is not clear if the standards are for teachers or students,

4. On page 4, first paragraph, it is stated: “Because previous studies didn’t directly…However, there was no reference to the studies.

5. On page 4, the research questions refer to the “degree of availability”. This is not defined. The research questions are not clear and do not seem to be aligned with the statement of the problem.

6. In the significance of the study, it was mentioned that methods of assessment of teachers could be beneficial for supervisors. However, does the study allow for this as assessment is not being done.

7. On page 5, the terms of the study are stated. This reads like a thesis, and not like a journal article.

8. The study was limited to a sample from Taif city. What is the rationale for selecting teachers in Taif, and not other cities, or even rural areas?

9. The NGSS website was mentioned several times. Why was the website used and not an official document?

10. The questionnaire that was used for the study (page 6), is it a new questionnaire designed for the study or one which is already available?

11. When interviews were used, it would be valuable to see the interview protocols and what was asked. It was also not clear how the interview data was analysed. Qualitative data needs to be shown with evidences such as transcripts of the interview. This seemed to be lacking.

12. The limitations of the study needs to be mentioned. This would enable suggestions for futre studies.

The article needs to be proofread to ensure the language is of quality

Reviewer #4: Congratulations, I have seen you have performed all the changes requested in the previous review. My only question is "where is the data availability statement?" I haven't been able to find it. I highly suggest you to use an open repository for your data such as OSF or similar to upload data from your research and making it available without restrictions. You should include this information in your manuscript as indicated by PLOS policies.

Reviewer #5: Title:

This title is vague. I could not identify the research objective through this title; is it an analysis of teachers' observations, an analysis of teacher programs, or just a survey study?

I think the study deals with the extent to which teachers adopt scientific and engineering practices in their teaching from their point of view.

Aligning means that the study analyzed teacher programs based on teachers' actual practices. Therefore, this study has nothing to do with aligning

Language:

All manuscript needs proof reading.

Abstract:

The method is not clear in the abstract.

Introduction:

There are no paragraphs related to the objective of the study

Questions:

The method is not clear in the abstract

Method:

Description of the instruments, the way of calculated the values related to the criterion of judgement such as intervals and therefore, this affected the assignment of the levels in all tables.

Results:

The criterion of judgement was as follows

degree of accessibility?

I think it is the level or degree of practice from their point of view.

Discussion:

The discussion is too short. You should use a separate section for the discussion and make it richer and more organized. Too short and does not do justice to the study's richness.

References:

Minor corrections are needed for the references.

Reviewer #6: Comment 1: I am sorry to say that this manuscript is confusing and, at times, was disjointed and unclear. For example, after thoroughly reading this manuscript more than once, I still do not know if the items of the questionnaire (indicators) were constructed by the author or taken from sources. Nevertheless, I hope my comments make more sense by explicitly stating my understanding of this work.

Comment 2: Simply stated, the author of this study, based on the NGSS website and several studies, identified 8 variables, each corresponding to a standard of teaching science to secondary school students. The author wrote (I assume) several Likert-scale statements (items) to, presumably, represent or address or measure each variable. Variables 1 through 8 are measured by 5, 6, 4, 7, 5, 7, 5, 5 items, respectively. Therefore, this is an instrument that measures 8 variables based on 44 Likert 5-point scale items. The author calls the variables "practices" and the items "indicators."

Comment 3: Then, an exploratory sample of 15 physics teachers was used to provide validity and reliability statistics. It is unclear if these teachers were also part of the study sample.

Comment 4: Then, this instrument was completed by 49 randomly selected secondary school teachers. It is not clear how they were randomly selected. How many teachers were asked, or how many questionnaires were sent? How many teachers said no, or what was the questionnaire return rate? The mean and standard deviation were calculated to represent each variable based on each variable's item scores. In addition, criteria based on the range of mean were defined: Very weak (<1.80), very weak (1.8-2.60), medium (2.60-3.40), high (3.40-4.20), very high (>4.20). The author did not provide justification for these cut-off scores.

Comment 5: Then, each teacher's total score on all 8 variables in this instrument was used to identify differences between teachers who hold graduate degrees and those who hold bachelor's degrees. A simple t-test of independent samples was used to declare significant differences between the two groups.

Comment 6: Then, the teachers were divided into three groups based on their reported years of experience (less than 5 years, 5-10 years, and more than 10 years), and a one-way ANOVA was used to compare the mean scores on this instrument of the three experience groups. No justification is given for grouping teachers' experience based on 5 years intervals. Are there other grouping intervals that might have produced significant differences? If a linear relationship exists between years of experience and scores on the instrument, examining the correlation between the two is more beneficial.

Comment 7: Finally, 6 teachers' supervisors were picked and interviewed to answer 8 open-ended questions about the presence of the scientific and engineering practices (variables) in the performance of the physics teachers. The supervisors' responses to the open-ended questions were compared with the teachers' responses on the instrument. Again, these supervisors were specifically selected, implying they know the teachers, which means no anonymity. Is this the case, and if so, why are these details not explicitly stated?

Comment 8: Clearly, this instrument is central to this study, and therefore, all the rules regarding test construction and statistical matrices apply. Each of the 8 variables/practices is a construct defined by the item loadings. However, these item loadings are hypothesized, not tested. While the author appears to have tried to provide the statistical rationale for using this instrument, it is far from sufficient. I believe that the major flaw in this study is the instrument itself. This instrument needs to be validated and its statistical properties identified. The first step is to administer this instrument to some 120 science teachers (using the 15 subjects per variable rule of thumb), conduct a thorough reliability and validity analysis, conduct factor analysis to determine item loading and appropriateness of items and variables, and revise as necessary. Only then may this instrument be used to make such inferences.

Comment 9: The structure and the presentation of the manuscript are much more consistent with what is normally required for a thesis or a dissertation than for research articles for publication. PLOS ONE provides what it deems to be acceptable structure and sequence. I recommend the following sequence: Introduction, Methods (procedure, participants, analysis), Results, Discussion, Limitations, and Conclusions.

Comment 10: There are too many punctuation and grammatical errors to list. Mixed tense in one paragraph is confusing (for example, 2nd paragraph on page 2 of the manuscript uses past tense at the beginning and then switches to present tense in the last sentence leaving the impression that the first several sentences are about elements that are no longer applicable, which is not the case). Also, serval sentences would benefit from re-writing them more clearly (for example, 2nd line on page 2, "teaching appeared such as…," 2nd paragraph on page 2, "need to adapt themselves to these new…." 3rd paragraph on page 2, "they encounter while studying science or in real life." And "i.e. whoever teaches science, in addition to skills of the engineer who solves problems." Another issue is the overuse of quotation marks (for example, 3rd paragraph on page 2, "can be described as "cognitive, discursive, and social activities" that can be …" What is so special about those words that dictate the use of quotation marks? If these words are special and verbatim from a reference, then a reference should immediately follow the quotation. I recommend proofreading this manuscript and using a professional copy editor.

Comment 11: Many paragraphs before methodology are unnecessary or can be trimmed to one or two sentences. For example, the study problem and the significance of the study can be easily combined, shortened, and embedded at the end of the introduction. The terms of the study can be summarized and written more concisely. There is no reason to justify using the qualitative method when one has qualitative data. The qualitative method used and how one interprets the finding are important, but not the justification unless it is unusual to use such methods.

Comment 12: Reading paragraph 4 on pages 2 and 3 gives the impression that the author constructed the eight variables (practices). But reading the terms of study scientific and engineering practices on page 4 reveals the author adopted these variables from the listed sources. This is confusing, unnecessary, repetitive, and internally inconsistent. Therefore, these two paragraphs need to be summarized and combined into one.

Comment 13: Everything before the methodology needs to be re-written under the introduction heading with more focus on what is being studied, why it is being studied, and what others have said about what is being studied. I believe the whole section needs to be well thought out and re-written.

Comment 14: The presentation of results needs improvements. The table should be formatted differently. The table columns should be mean (SD), degree of presence should include the range of the mean and the designation, and rank.

Comment 15: It is unclear why a table for each practice (variable) is shown. Such a presentation might be appropriate if the work addresses the validation and the statistical properties of an instrument, but not when the work is focused on the mean of each variable (practice). Table 1 would be sufficient for this type of work where means are compared.

Comment 16: Also, the author does not seem to distinguish between results and discussion. The results section should be devoted to what the author finds. The discussion section should discuss the results and compare them to what others have found.

Comment 17: The limitations section needs to be well thought out and more clearly written and articulated.

Reviewer #7: 1. Introduction and literature review : first paragraph (1-4) who? “ explained that education ….), you need to write (1-4) at the end of the sentence.

2. The study limits section to be transferred to the end of discussion

3. The results and discussion parts are mixed.

4. Table 3 showed that general mean stated as weak while in the text stated as medium.

Reviewer #8: (No Response)

7. PLOS authors have the option to publish the peer review history of their article (what does this mean?). If published, this will include your full peer review and any attached files.

Reviewer #2: No

Reviewer #3: No

Reviewer #4: No

Reviewer #5: No

Reviewer #6: **Yes: **Dr. Ali M. AL-Asadi

Reviewer #7: No

Reviewer #8: No

---

## [Author Response · Author response to Decision Letter 1]

15 May 2022

Reviewer # 3

1. Abstract does not indicate the need for the study 

Modified.

2. On page 2. The problem statement is not clear. You have mentioned that science teachers of all levels need ………….

The problem is clear

3. On page 3, you have stated the practices in NGSS………….

It has been shown that teachers' practices contribute to the development of standards in students

4. On page 4, first paragraph, it is stated: “Because previous studies …………..

Modified

5. On page 4, the research questions refer to the “degree of availability”. This is not defined. The research ……………

Modified

6. In the significance of the study, it was mentioned that methods of assessment of teachers could be beneficial for …………..

The importance section has been removed as requested by reviewers

7. On page 5, the terms of the study are stated. This reads like a thesis, and not like a journal article.

It has not been deleted, due to the importance of defining the study terms

8. The study was limited to a sample from Taif city. What is the rationale for selecting teachers in Taif, ………..

Because the study focuses on physics teachers in the city of Taif only

9. The NGSS website was mentioned several times. Why was the website used and not an official document?

Because the website is the primary reference for NGSS standards 

10. The questionnaire that was used for the study (page 6), is it a new questionnaire designed for the study or one which is already available?

The questionnaire was built by the researcher after referring to the references and studies

11. When interviews were used, it would be valuable to see the interview protocols and what was asked………….

This is explained in the construction and application procedure of the interview

12. The limitations of the study needs to be mentioned. This would enable suggestions for futre studies.

Modified

13. The article needs to be proofread to ensure the language is of quality

The manuscript was revised by an English language specialist

Reviewer # 5

 Title : This title is vague. I could not identify the research objective through this title; is it an analysis of teachers' observations…………… 

The study attempted to reveal the degree of availability of scientific and engineering practices aligning with NGSS standards in the performance of teachers

Language: All manuscript needs proof reading. 

The manuscript was revised by an English language specialist

Abstract: The method is not clear in the abstract. 

Modified

Introduction: There are no paragraphs related to the objective of the study 

Modified

Questions: The method is not clear in the abstract 

Modified

Method: Description of the instruments, the way of calculated the values …………. 

Modified

Results: The criterion of judgement was as follows degree of accessibility? ………

Yes

Discussion: The discussion is too short. You should use a separate section for the discussion and make it richer and more organized. ………. 

Since the manuscript presented the results of each scientific and engineering practice separately, this requires the presentation and discussion of each result directly.

References: Minor corrections are needed for the references.

Modified

Reviewer # 6

Comment 1: I am sorry to say that this manuscript is confusing and, at times, was disjointed and unclear. ……………..

The questionnaire was built by the researcher after referring to the references and studies

Comment 2: Simply stated, the author of this study, based on the NGSS website and several studies, identified 8 variabl……

ok

Comment 3: Then, an exploratory sample of 15 physics teachers was used to provide validity and reliability statistics………….

It was clarified that the exploratory sample was outside the study sample

Comment 4: Then, this instrument was completed by 49 randomly selected ……………….

Modified

Comment 6: Then, the teachers were divided into three groups based on their reported years of experience …………….

This classification was adopted after referring to several studies that used it

Comment 7: Finally, 6 teachers' supervisors were picked and interviewed to …………...

This was clarified, and that supervisors give their opinions based on their actual follow-up of these teachers

Comment 8: Clearly, this instrument is central to this study, and therefore, all the rules regarding test construction and statistical matrices apply.…..

The characteristics of the psychometric instruments have been confirmed

Comment 9: The structure and the presentation of the manuscript are much more consistent with what is ………..

Modified

Comment 11: Many paragraphs before methodology are unnecessary or can be trimmed to one or two sentences. For example, the study problem ………………..

Modified

Comment 12: Reading paragraph 4 on pages 2 and 3 gives the impression that the author constructed …………….

Modified

Comment 13: Everything before the methodology needs to be re-written under the introduction …………..

Modified

Comment 14: The presentation of results needs improvements. The table should be formatted differently……….

Modified

Comment 15: It is unclear why a table for each practice (variable) is shown. Such a presentation might …………..

Since scientific and engineering practices are the focus of this study, it is better to present each practice in a separate table, as well as discuss it independently

Comment 16: Also, the author does not seem to distinguish ……………

Since the manuscript presented the results of each scientific and engineering practice separately, this requires the presentation and discussion of each result directly.

Comment 17: The limitations section needs to be well thought out and more clearly written and articulated

Modified

Reviewer # 7

1. Introduction and literature review : first paragraph (1-4) who? “ explained that education ….)…………..

Modified

2. The study limits section to be transferred to the end of discussion

Modified

3. The results and discussion parts are mixed.

Yes

4. Table 3 showed that general mean stated as weak while in the text stated as medium.

Modified

Reviewer # 8

Point 1: In the (Abstract): researcher confirmed that there are two aims for this study which were ………………….

In the literature of the study, all the aims of the study were addressed

Point 2: under the (Study Problem): (…However, …………..

modified

Point 3: The whole section of (Significance of the Study) needs to be deleted, and no need for it in the structure of writing this article. 

modified

Point 4: The whole section of (Terms of the Study) needs to be deleted and some the information in relation to the definitions just need to be added to the literature review.

It has not been deleted, due to the importance of defining the study terms

Point 5: The whole section of (Study Limits and Limitations) needs to be moved to different parts in this…………

modified

Point 6: under the (Methodology): researcher did not specify which one of the mixed methods..........

It was mentioned that the questionnaire was applied to the teachers first, then the interview to the supervisors 

Point 7: under the (Study Instruments): the researcher stated that (….. After determining the ……………..

The objectives of the questionnaire are the same as the objectives of the study, and they were predetermined

Point 8: In terms of the validation of the questionnaire, researcher applied the internal consistency ………..

A doctor specializing in educational statistics was consulted, and he stated that the internal consistency is very appropriate

Point 9: Under the (Study Result): all the results need to be separated from the discussion and need to be ………..

Since the manuscript presented the results of each scientific and engineering practice separately, this requires the presentation and discussion of each result directly.

Point 10: Under the (Conclusion): Conclusion needs to be rewritten again. it should be a summary for only ………...

modified

Finally, there is a need to bring uniformity in the reference section.

Verified

---

## [Decision Letter · Decision Letter 2]

17 Jun 2022

PONE-D-21-34137R2Scientific and engineering practices aligned with the NGSS in the performance of secondary stage physics teachersPLOS ONE

Dear Dr. Alsalamat,

Thank you for submitting your manuscript to PLOS ONE. After careful consideration, we feel that it has merit but does not fully meet PLOS ONE’s publication criteria as it currently stands. Therefore, we invite you to submit a revised version of the manuscript that addresses the points raised during the review process.

We look forward to receiving your revised manuscript.

Kind regards,

Yiming Tang, Ph.D.

Academic Editor

PLOS ONE

Reviewers' comments:

Reviewer's Responses to Questions

**Comments to the Author**

1. If the authors have adequately addressed your comments raised in a previous round of review and you feel that this manuscript is now acceptable for publication, you may indicate that here to bypass the “Comments to the Author” section, enter your conflict of interest statement in the “Confidential to Editor” section, and submit your "Accept" recommendation.

Reviewer #4: All comments have been addressed

Reviewer #6: (No Response)

Reviewer #7: (No Response)

Reviewer #8: (No Response)

2. Is the manuscript technically sound, and do the data support the conclusions?

Reviewer #4: Yes

Reviewer #6: No

Reviewer #7: Yes

Reviewer #8: Partly

3. Has the statistical analysis been performed appropriately and rigorously? 

Reviewer #4: Yes

Reviewer #6: No

Reviewer #7: I Don't Know

Reviewer #8: Yes

4. Have the authors made all data underlying the findings in their manuscript fully available?

Reviewer #4: No

Reviewer #6: Yes

Reviewer #7: Yes

Reviewer #8: (No Response)

5. Is the manuscript presented in an intelligible fashion and written in standard English?

Reviewer #4: Yes

Reviewer #6: Yes

Reviewer #7: No

Reviewer #8: Yes

6. Review Comments to the Author

Reviewer #4: Everything is ok. I see you have addressed the previous comments suggested by the reviewers. The only concern I have is your data availability statement. You indicate that "data will be available on request", but that is not enough. There are lots of free data repositories such as OSF that allow you to share your data with the world. Please upload your data to a repository and share the link before publication.

Reviewer #6: I am sorry that the author has not adequately addressed my comments. Although the author indicated that some sentences were modified, the modifications are insufficient. I do not believe that the central issue of establishing the author-constructed instrument's reliability and the validity has been conducted properly. For example, stating that "The characteristics of the psychometric instruments have been confirmed" is insufficient to address this matter. Also, the author's justification that "Since the manuscript presented the results of each scientific and engineering practice separately, this requires the presentation and discussion of each result directly" is insufficient.

Overall, there need to be significant changes to the content, the style, and the presentation. Unfortunately, the manuscript still reads like an abbreviated thesis, and the concerns I raised have not been thoroughly or adequately addressed.

Thank you.

Reviewer #7: The first paragraph in introduction need punctuation, the paper need language editing before publishing.

the result and discussion sections should be separated and not mixed

Reviewer #8: For the manuscript titled " Scientific and engineering practices aligned with the NGSS in the performance of secondary stage physics teachers ", there are some points that were not addressed and need to be revised; especially, in relation to the literature of the study, research methodology and conclusion as following:

Point 1: In terms of the aims of the study that were mentioned in the abstract. the literature of this study still has not covered all these aims and questions, and there are no modifications that were made in the literature.

Point 2: under the (Methodology): researcher still did not specify which one of the mixed methods approaches was applied in this study. There is no mention which one did the researcher start with it before the other (questionnaire or interviews)?. And what the kind of mixed methods that was applied in this study Sequential Explanatory Design or Sequential Exploratory Design.

Point 3: Under the (Study Limits and Limitations), researcher did not clarify which one of the three major dimensions of the NGSS was chosen and why. In addition, researcher did not explain or mention about these three major dimensions in the literature review.

Finally: Under the (Conclusion): there were no modifications that were made by researcher. Conclusion needs to be rewritten again. it should be a summary for only the ideas that have strong finial impression. Also, all the limitations under the conclusion need to be moved to the section that was moved by the researcher before the conclusion. In addition, all the information from (…the results revealed that the teachers did...) needs to be placed under new section called Implications of this study.

7. PLOS authors have the option to publish the peer review history of their article (what does this mean?). If published, this will include your full peer review and any attached files.

Reviewer #4: No

Reviewer #6: **Yes: **Ali M. AL-Asadi

Reviewer #7: No

Reviewer #8: No

---

## [Author Response · Author response to Decision Letter 2]

25 Jul 2022

Reviewer # 6

- I am sorry that the author has not adequately addressed………….

 Response: Modified: The results have been separated from the discussion

-Overall, there need to be significant ……………

Response: Modified: The results have been separated from the discussion

Reviewer # 7

- The first paragraph in introduction need punctuation, the paper need language editing before publishing.

Response: The study was proofread by a specialized proofreader

- the result and discussion sections should be separated and not mixe

Response: Modified: The results have been separated from the discussion

Reviewer # 8

Point 1: In terms of the aims of the study that were mentioned in the abstract…… 

Response: Modified

Point 2: under the (Methodology): researcher still …………….

Response: Modified

Point 3: Under the (Study Limits and Limitations), researcher did not clarify which one of the three major dimensions of the NGSS ……….

Response: Modified

Finally: Under the (Conclusion): there were no modifications that were made by ……….

Response: Modified

---

## [Decision Letter · Decision Letter 3]

11 Aug 2022

PONE-D-21-34137R3Scientific and engineering practices aligned with the NGSS in the performance of secondary stage physics teachersPLOS ONE

Dear Dr. Alsalamat,

Thank you for submitting your manuscript to PLOS ONE. After careful consideration, we feel that it has merit but does not fully meet PLOS ONE’s publication criteria as it currently stands. Therefore, we invite you to submit a revised version of the manuscript that addresses the points raised during the review process.

We look forward to receiving your revised manuscript.

Kind regards,

Yiming Tang, Ph.D.

Academic Editor

PLOS ONE

Reviewers' comments:

Reviewer's Responses to Questions

**Comments to the Author**

1. If the authors have adequately addressed your comments raised in a previous round of review and you feel that this manuscript is now acceptable for publication, you may indicate that here to bypass the “Comments to the Author” section, enter your conflict of interest statement in the “Confidential to Editor” section, and submit your "Accept" recommendation.

Reviewer #4: All comments have been addressed

Reviewer #6: (No Response)

Reviewer #8: (No Response)

2. Is the manuscript technically sound, and do the data support the conclusions?

Reviewer #4: Yes

Reviewer #6: Partly

Reviewer #8: Yes

3. Has the statistical analysis been performed appropriately and rigorously? 

Reviewer #4: Yes

Reviewer #6: No

Reviewer #8: I Don't Know

4. Have the authors made all data underlying the findings in their manuscript fully available?

Reviewer #4: Yes

Reviewer #6: Yes

Reviewer #8: Yes

5. Is the manuscript presented in an intelligible fashion and written in standard English?

Reviewer #4: Yes

Reviewer #6: Yes

Reviewer #8: Yes

6. Review Comments to the Author

Reviewer #4: Congratulations, the manuscript looks suitable for publication to me now. I am happy to see you have performed the changes that were suggested in the previous reviews.

Reviewer #6: I will start by thanking the author for making the changes and reorganizing the manuscript. It reads much better now. However, I still have one major concern, which I shall outline in detail.

My analysis, critique, and review are based on my understanding of this study. Therefore, briefly stating my understanding is vital to understanding my critique.

1. The author identified eight scientific and Engineering practices based on NGSS: asking questions, developing and using models, planning and investigating, analyzing and interpreting data, involvement with proofs and evidence, interpretations and solutions design, obtaining, evaluating and communicating information, and using mathematics and computational thinking.

1. Then, the author devised a questionnaire comprising 44 indicators that measure these eight practices on a Likert’s 5-point scale (number of indicators per practice ranged from 4 to 7).

2. The questionnaire was then administered to a sample of 15 teachers, and based on their scores, internal consistency and Cronbach’s Alpha coefficients were calculated to “verify” the validity and reliability of the instrument.

3. Next, the author administered the questionnaire to 49 secondary school physics teachers to identify the degree to which the performance of physics teachers is aligned with NGSS scientific and engineering practices and if teachers’ qualifications and years of experience affect the degree to which teachers’ performance and the NGSS practices are aligned.

If this is correct, then basically, the author is trying to do two things at the same time: (1) constructing an instrument (questionnaire) that measures the eight practices and (2) using this instrument to (a) measure the degree to which the teaching is aligned with NGSS practices and (b) detect differences in adherence to NGSS practices based on qualifications and years of experience.

4. Firstly, the eight practices and the 44 indicators are not separate dimensions. They are the same in that the 44 indicators reflect (or measure) the eight practices. Therefore, statements such as, “The questionnaire comprised (8) practices and (44) indicators” are inaccurate and should read “The questionnaire comprised of 44 indicators representing (or measuring) eight practices.” Also, bracketing the number of practices and indicators is confusing and unnecessary.

5. Secondly and most importantly, the issue of reliability and validity. As I have indicated in my first review of this manuscript, a serious issue exists in establishing the validity and reliability of this instrument. It does not matter whether it is a questionnaire or an instrument; it is still a test that must go through the proper test construction process, including adequate validity and reliability procedures. There is no need to establish reliability and validity when constructing a questionnaire that asks for the sex and age of participants and their preference for a flavour of ice cream. But, in a case where multiple dimensions (or factors) define a construct (or constructs), establishing reliability and validity properly is of utmost importance, even in low-stakes testing.

6. Validity reflects the instrument’s ability to measure what it purports to measure. Reliability reflects the consistency of the instrument’s measurements. An instrument cannot be valid unless it is reliable, but a reliable instrument may not be valid. The author provided a range of internal consistency and Cronbach’s Alpha based on a very small sample of 15 teachers. The sample size is simply too small to establish reliability and validity for an instrument comprising 44 items measuring eight dimensions. Moreover, the small sample size notwithstanding, these two statistics may show that the instrument is reliable, but there is nothing about its validity.

7. Therefore, in the final analysis, the first part of this study is related to test construction, which was not conducted properly. The second part of the study is procedurally acceptable, but its results are based on an instrument that lacks validity and reliability, or at least its reliability and validity are questionable.

8. It is rather strange that this significant limitation is not indicated and included in the limitation section. Furthermore, the limitation section is relatively thin.

9. The second paragraph of the conclusion section repeats what is briefly mentioned in the limitation. The entire paragraph should be moved to the limitation section.

Obviously, a lot of work has gone into building this instrument, and it may be made a standard instrument to test adherence to NGSS practices providing proper validity and reliability studies. I suggest the author continues this work and develops this instrument to its full potential.

Whether to publish this manuscript or not is in the hands of the journal editors. To adequately address my concern, the test construction process needs to be followed. Data must be collected on a larger sample, followed by proper reliability and validity analyses. Alternatively, explicit statements must be made to emphasize this instrument's lack of (or questionable) reliability and validity and the caution and limitation that must be exercised when interpreting these results.

Reviewer #8: For the manuscript titled " Scientific and engineering practices aligned with the NGSS in the performance of secondary stage physics teachers ", there still some limitations under the section of “Conclusion”. All these limitations under the conclusion need to be moved to the “study limits and limitations” and make them concise.

7. PLOS authors have the option to publish the peer review history of their article (what does this mean?). If published, this will include your full peer review and any attached files.

Reviewer #4: No

Reviewer #6: **Yes: **Dr. Ali M. AL-Asadi

Reviewer #8: No

---

## [Author Response · Author response to Decision Letter 3]

5 Sep 2022

Reviewer #6: 

- Firstly, the eight practices and the 44 indicators are not separate dimensions………………

Response:Modified

- Secondly and most importantly, the issue of reliability and validity. As I have indicated in my first review of this manuscript, a serious issue exists in establishing the validity and reliability of this instrument ……….

Response:Modified

Reviewer #8: 

- For the manuscript titled " Scientific and engineering practices aligned with the NGSS in the performance of secondary stage physics teachers ", there still some limitations under the section of “Conclusion”. All these limitations under the conclusion need to be moved to the “study limits and limitations” and make them concise.

Response:Modified

---

## [Decision Letter · Decision Letter 4]

13 Sep 2022

Scientific and engineering practices aligned with the NGSS in the performance of secondary stage physics teachers

PONE-D-21-34137R4

Dear Dr. Alsalamat,

We’re pleased to inform you that your manuscript has been judged scientifically suitable for publication and will be formally accepted for publication once it meets all outstanding technical requirements.

Kind regards,

Yiming Tang, Ph.D.

Academic Editor

PLOS ONE

Additional Editor Comments (optional):

Reviewers' comments:

Reviewer's Responses to Questions

**Comments to the Author**

1. If the authors have adequately addressed your comments raised in a previous round of review and you feel that this manuscript is now acceptable for publication, you may indicate that here to bypass the “Comments to the Author” section, enter your conflict of interest statement in the “Confidential to Editor” section, and submit your "Accept" recommendation.

Reviewer #6: All comments have been addressed

Reviewer #8: All comments have been addressed

2. Is the manuscript technically sound, and do the data support the conclusions?

Reviewer #6: Yes

Reviewer #8: Yes

3. Has the statistical analysis been performed appropriately and rigorously? 

Reviewer #6: Yes

Reviewer #8: N/A

4. Have the authors made all data underlying the findings in their manuscript fully available?

Reviewer #6: Yes

Reviewer #8: Yes

5. Is the manuscript presented in an intelligible fashion and written in standard English?

Reviewer #6: Yes

Reviewer #8: Yes

6. Review Comments to the Author

Reviewer #6: (No Response)

Reviewer #8: (No Response)

7. PLOS authors have the option to publish the peer review history of their article (what does this mean?). If published, this will include your full peer review and any attached files.

Reviewer #6: **Yes: **Dr. Ali M. AL-Asadi

Reviewer #8: No

---

## [Editor Report · Acceptance letter]

29 Sep 2022

PONE-D-21-34137R4 

Scientific and engineering practices aligned with the NGSS in the performance of secondary stage physics teachers 

Dear Dr. Alsalamat:

I'm pleased to inform you that your manuscript has been deemed suitable for publication in PLOS ONE. Congratulations! Your manuscript is now with our production department. 

Kind regards, 

on behalf of

Professor Yiming Tang 

Academic Editor

PLOS ONE